# Adipose-Derived Mesenchymal Stem Cells (ADSCs) Have Anti-Fibrotic Effects on Lung Fibroblasts from Idiopathic Pulmonary Fibrosis (IPF) Patients

**DOI:** 10.3390/cells13242050

**Published:** 2024-12-12

**Authors:** Noriko Ouji-Sageshima, Aiko Hiyama, Makiko Kumamoto, Masahiro Kitabatake, Atsushi Hara, Ryutaro Furukawa, Shigeto Hontsu, Takeshi Kawaguchi, Noriyoshi Sawabata, Shigeo Muro, Toshihiro Ito

**Affiliations:** 1Department of Immunology, Nara Medical University, Kashihara 634-8521, Nara, Japan; osage@naramed-u.ac.jp (N.O.-S.); aiko.rp.gp21@gmail.com (A.H.); mkita@naramed-u.ac.jp (M.K.); atsush.har@gmail.com (A.H.); rfurukawa-tky@naramed-u.ac.jp (R.F.); 2Department of Respiratory Medicine, Nara Medical University, Kashihara 634-8521, Nara, Japan; kumamoto.makiko.ze@mail.hosp.go.jp (M.K.); hontsu@naramed-u.ac.jp (S.H.); smuro@naramed-u.ac.jp (S.M.); 3Department of Thoracic and Cardiovascular Surgery, Nara Medical University, Kashihara 634-8521, Nara, Japan; kawagu@naramed-u.ac.jp (T.K.); nsawabata@naramed-u.ac.jp (N.S.)

**Keywords:** IPF, ADSC, collagen, PTPRR, humanized mice

## Abstract

Idiopathic pulmonary fibrosis (IPF) is the most common type of fibrosis in lungs, characterized as a chronic and progressive interstitial lung disease involving pathological findings of fibrosis with a median survival of 3 years. Despite the knowledge accumulated regarding IPF from basic and clinical research, an effective medical therapy for the condition remains to be established. Thus, it is necessary for further research, including stem cell therapy, which will provide new insights into and expectations for IPF treatment. Recently, it has been reported that one of the new therapeutic candidates for IPF is adipose-derived mesenchymal stem cells (ADSCs), which have several benefits, such as easy accessibility and minimal morbidity compared to bone marrow-derived mesenchymal stem cells. Therefore, we investigated the possibility of ADSCs as a therapeutic candidate for IPF. Using human lung fibroblasts (LFs) from IPF patients, we demonstrated that human IPF LFs cocultured with ADSCs led to reduced fibrosis-related genes. Further analysis revealed that ADSCs prevented the activation of the ERK signaling pathway in IPF LFs via the upregulation of protein tyrosine phosphatase receptor-type R (PTPRR), which negatively regulates the ERK signaling pathway. Moreover, we demonstrated that intravascular administration of ADSCs improved the pathogenesis of bleomycin-induced pulmonary fibrosis with reduced collagen deposition in histology and hydroxyproline quantification and collagen markers such as the gene expression of types I and III collagen and α-smooth muscle actin (α-SMA) in a murine model. ADSC transfer was also investigated in a humanized mouse model of lung fibrosis induced via the infusion of human IPF LFs, because the bleomycin installation model does not fully recapitulate the pathogenesis of IPF. Using the humanized mouse model, we found that intravascular administration of ADSCs also improved fibrotic changes in the lungs. These findings suggest that ADSCs are a promising therapeutic candidate for IPF.

## 1. Introduction

Idiopathic pulmonary fibrosis (IPF) is a chronic, progressive, and irreversible interstitial pneumonia caused by the excessive production and deposition of extracellular matrix components, including activated fibroblasts and α-smooth muscle actin (α-SMA)-expressing myofibroblasts [1,2]. Currently, anti-fibrotic agents, pirfenidone and nintedanib, are used for the treatment of IPF; however, neither of them can cure the disease by themselves [3,4]. In fact, IPF is still ultimately fatal, with accumulating data suggesting a median survival of approximately 3 years from the time of diagnosis [5]. Thus, it is urgently necessary to develop new and effective therapies for IPF.

Recent research regarding mesenchymal stem cells (MSCs) derived from various somatic tissues, including bone marrow (BM), cord blood, and adipose tissue, has been performed in the fields of regenerative and medical treatment [6]. In addition, MSCs, also known as multipotent cells, have been reported to prevent the development of tissue fibrosis in animal experimental models and preclinical studies. In particular, in the context of clinical applications, MSCs from fat tissue, adipose-derived mesenchymal stem cells (ADSCs), are an important candidate because of their high accessibility with minimal invasiveness and fewer ethical limitations [7]. Importantly, adipose tissue contains higher densities of MSCs compared with BM [8]. Besides the fact that ADSCs are heterogeneous [9], they produce a larger number of growth factors than BM-derived MSCs for tissue regeneration and repair [10] and exhibit immunomodulatory properties and differentiation abilities similar or superior to BM-derived MSCs [11,12]. However, little is known about the precise mechanism for the favorable effect of ADSCs on pulmonary fibrosis (PF), especially on lung fibroblasts (LFs) from IPF patients.

The development of PF is most commonly studied in animal models using bleomycin (BLM). Notably, a BLM-induced PF model does not generally result in the long-term histological characteristics seen in patients with IPF [13]. Recent attempts have been made to develop an irreversible PF by “humanizing” IPF models. The best-characterized humanized model involves the intravenous administration of human IPF fibroblasts to mice with severe immunodeficiency [14].

In this study, we examined the anti-fibrotic effect of ADSCs on LFs from IPF patients and its mechanism and also investigated the therapeutic effect of intravenous ADSC administration on not only a BLM-induced PF model, but also on a humanized IPF mouse model.

## 2. Materials and Methods

### 2.1. Human Lung Samples

Human lung samples were obtained from lung cancer patients with IPF who underwent surgical excision between August 2015 and December 2019 at Nara Medical University Hospital. All patients who participated in this study provided informed consent. We obtained IPF patient-derived LFs from both fibrotic (F) and non-fibrotic (NF) areas. The NF and F areas, which did not contain the lung cancer in excised lung samples, were decided under thin-section chest computed tomography by two independent specialists in respirology. The diagnosis of IPF followed an official ATS/ERS/JRS/ALAT clinical **practice** guideline. LFs were isolated from both F and NF areas of human lung samples according to a previously described method [15,16]. Briefly, the NF-area and F-area lung samples obtained from the patients were minced and transferred to six-well tissue culture plates, and cultured with 15% FCS-DMEM including 1% penicillin/streptomycin and 0.1% amphotericin B for 1 week, and non-adherent cells were removed, including red cells. After 1 week of culture, the major population of adherent cells was found to be fibroblasts, which were then transferred to a 75 cm^2^ culture flask. We continued to culture the LFs at least three times and then stored the cells at −80 °C. The characterization of the isolated LFs is shown in Figure 1. We confirmed that all LFs expressed CD90 and CD140a (fibroblast markers), and did not express other markers such as CD45 (a leukocyte marker), CD31, or CD144 (endothelial markers).

We used the cryopreserved and passaged LFs for the coculture system for both ADSCs and LFs.

All experiments involving human samples in this study were approved by the Ethics Committee of Nara Medical University.

### 2.2. Cell Preparation

Human ADSCs (hADSCs) were purchased from LONZA Co., Ltd. (Basel, Switzerland), and the cells were expanded using a serum-free medium (Rohto Pharmaceutical Co., Ltd., Osaka, Japan).

### 2.3. Coculture System for Both ADSCs and LFs

LFs were seeded at 5 × 10^4^ cells/well in a 24-well plate. After incubation overnight, 2 × 10^5^ ADSCs were incubated onto a 1 µm pore cell chamber (Transwell^®^; Thermo Fisher Scientific, Waltham, MA, USA) and indirectly cocultured with LFs. For HGF stimulation, the LFs were incubated with 1 µg/mL of HGF (PeproTech^®^; Thermo Fisher Scientific, Waltham, MA, USA). After 24 h of incubation, the LFs were collected for RNA analysis and Western blot analysis.

### 2.4. Overexpression of PTPRR Gene in LFs

The plasmid of the *PTPRR* gene with the FLAG-tag sequence in the C terminal (#RC212896) was purchased from OriGene Technologies, Inc. (Rockville, MD, USA) and amplified in *E. coli* JM109 Competent Cells (Takara Bio Inc., Kusatsu, Japan), and then the *PTPRR* plasmid was purified using NucleoSpin Pure Plasmid^®^ (Takara Bio Inc., Kusatsu, Japan). Sequencing was performed on the purified *PTPRR* plasmid to check the gene sequence. The concentration of DNA of the *PTPRR* plasmid was determined based on optical density. The transfection of the *PTPRR* gene into the LFs was performed via electroporation using Nucleofector^®^ (Lonza Co., Ltd., Basel, Switzerland) according to the manufacturer’s protocol.

### 2.5. Mouse PF Model

We used two kinds of PF models: a BLM-induced PF model and an IPF-humanized model. For the BLM-induced PF model, C57BL/6N mice (7 weeks old) were purchased from CLEA Japan Inc. (Higashiyama, Japan). After habituation for one week, BLM (1.5 mg/kg) or PBS 50 µL/body (for control) were administered intratracheally (i.t.) to the mice. On day 3 after BLM administration, cryopreserved ADSCs (1.5 × 10^6^ cells/body) or PBS 200 µL/body (for control) were administered intravenously, and the BLM-induced PF model mice were euthanized on day 14 after BLM administration for analysis. For the IPF-humanized model, severe combined immunodeficiency (SCID-Beige) mice (7 weeks old) were purchased from Oriental BioService Inc. (Kyoto, Japan). After habituation for one week, 1.5 × 10^6^ cells/body of LFs derived from the F-area were administrated intravenously. On day 35 after LF administration, cryopreserved ADSCs (1.5 × 10^6^ cells/body) or PBS 200 µL/body (for control) were administered intravenously, and then the IPF-humanized model mice were euthanized on day 63 after F-LF administration for analysis. The Animal Care and Use Committee at Nara Medical University approved all animal experiments conducted in this study, and all methods were carried out based on the Policy on the Care and Use of Laboratory Animals, Nara Medical University and Animal Research: Reporting of In Vivo Experiments (ARRIVE) guidelines. The use of LFs and hADSCs was approved by the Nara Medical University Ethics Committee.

### 2.6. Pathological Analysis

For the analysis of the mouse PF model, 4% paraformaldehyde-fixed paraffin-embedded left-lobe lung tissue was used for pathologic analysis, HE staining, and Masson’s trichrome staining. Fibrosis was assessed by 3 independent pathologists according to the Ashcroft score [17]. The collagen content in 1 of 4 right lobes of the lung was used for an additional assessment of fibrosis on a mouse PF model. The collagen analysis was performed using a Sircol Collagen Assay Kit (Biocolor Ltd., Carrickfergus, UK) using the 3rd right lobe.

### 2.7. RNA Analysis (Quantitative PCR (qPCR))

For the RNA analysis of the in vitro experiment, total RNA was extracted from the cells using a RNeasy mini Kit (Qiagen, Hilden, Germany); then, RT-PCR was performed using a TaqMan assay (Thermo Fisher Scientific, Waltham, MA, USA). The expression levels were analyzed via the ΔΔCT method normalized with *GAPDH* expression. We performed the TaqMan gene expression assay using the following TaqMan probe: *GAPDH* (Hs03929097), *ACTA2* (Hs00426835), *COL1A2* (Hs01028956), *COL3A1* (Hs00943809), *PTPRR* (Hs00373136). For the RNA analysis of mouse PF models, 1 of 4 right lobes of the lung was stored in RNAlater^®^ Stabilization Solution (Thermo Fisher Scientific, Waltham, MA, USA) at −80 °C until RNA extraction. The total RNA was isolated using NucleoSpin^®^ RNA (MACHEREY-NAGEL GmbH & Co., KG, Düren, Germany); then, RT-PCR was performed via TaqMan assay (Thermo Fisher Scientific, Waltham, MA, USA). The expression levels were analyzed with the ΔΔCT method using the normalized value on *Gapdh* expression. We performed the TaqMan gene expression assay using the following TaqMan probe: *Gapdh* (Mm99999915), *Acta2* (Mm00808218), *Col1a2* (Mm01309565), *Col3a1* (Mm00802300), and *Ptprr* (Mm00448522).

### 2.8. Microarray Analysis

Total RNA was extracted from the LFs with or without hADSC treatment using an RNeasy mini Kit (Qiagen, Hilden, Germany). DNA microarray analysis was carried out by DNA Chip Research Inc. (Kawasaki, Japan) using a SurePrint G3 Human GE Microarray 8 × 60 K (version 3.0, Agilent Technologies, Santa Clara, CA, USA). The detailed procedure is described elsewhere [15]. Briefly, Total RNA was amplified and labeled with cyanine 3, and then reverse-transcribed into double-stranded cDNA. Quantification of the intensity values for each scanned feature was carried out by Feature Extraction Software (version 10.7.3.1, Agilent), and then values were normalized with GeneSpring GX (version 11.0.2, Agilent).

### 2.9. Western Blotting for Protein Analysis

Protein analysis was performed using Western blotting or the WES system (ProtineSimple^TM^, Bio-techne, Minneapolis, MN, USA). The LFs after stimulation or the transfection of *PTPRR* were lysed via RIPA buffer with a protein inhibitor cocktail (WAKO, Osaka, Japan) and then centrifuged at 12,000 rpm for 15 min at 4 °C. The supernatant was used for protein analysis. The expression levels of each protein were normalized for GAPDH, β-actin, and total (t)-ERK expression, respectively. The antibodies used in the protein analysis were as follows: phosphorylated (p)-ERK (#9106, Cell Signaling Technology Inc., Danvers, MA, USA), t-ERK (#9102, Cell Signaling Technology Inc., Danvers, MA, USA), Collagen Type I Antibody (600-401-103, Rockland Immunochemicals, Inc., Limerick, PA), anti α-SMA monoclonal antibody (clone 1A4, Abcam, Cambridge, UK), GAPDH (#5174, Cell Signaling Technology Inc., Danvers, MA, USA), anti-FLAG M2 antibody (F3165, Sigma-Aldrich, St. Louis, MO, USA), and anti-β-actin (A5441, Sigma-Aldrich, St. Louis, MO, USA).

### 2.10. Flow Cytometry

The isolated LFs were characterized using flow cytometry. Ater trypsinization, the LFs were washed three times with PBS containing BSA at 1%, and then these cells were used for flow cytometry. The antibodies used in the flow cytometry were as follows: BV421-labeled anti CD90 monoclonal antibody, FITC-labeled anti CD45 monoclonal antibody, PE-labeled anti CD144 monoclonal antibody, PE-Cy7-labeled anti CD140a monoclonal antibody, and Alexa Fluor-647-labeled anti CD31 (BioLegend, San Diego, CA, USA).

### 2.11. Statistical Analysis

Statistical significance was assessed by Student’s *t*-test or one-way analysis of variance (ANOVA). Less than 0.05 in *p*-values was considered a significant difference. Statistical analyses were carried out by GraphPad Prism 4.0 software (GraphPad Software, Boston, MA, USA). All data are shown as mean ± SEM from at least two independent experiments

## 3. Results

### 3.1. ADSCs Have Anti-Fibrotic Effects on IPF Patient-Derived LFs

We obtained IPF patient-derived LFs from both the F area and NF area to analyze the therapeutic effect, excluding the influence of genetic background [15]. To investigate whether ADSCs have anti-fibrotic effects on LFs, we performed cocultures on LFs with or without ADSCs separately using a Boyden chamber (Figure 2A) and analyzed the gene expression of fibrotic markers such as *ACTA2*, *COL1A2*, and *COL3A1* of LFs from both the F area (F-LFs) and NF area (NF-LFs). The expressions of *ACTA2*, *COL1A2*, and *COL3A1* in both F-LFs and NF-LFs were significantly inhibited by ADSC treatment (Figure 2B). Additionally, a protein analysis showed that the expression of both α-SMA and collagen type I in LFs was reduced by treatment with ADSCs (Figure 2C). These results suggest that ADSCs have an anti-fibrotic effect for both F-LFs and NF-LFs.

### 3.2. ADSC Treatment Upregulates the Expression of PTPRR in LFs

Next, a DNA microarray analysis was performed to identify which genes were correlated with anti-fibrotic processes using ADSC treatment. We compared the gene expression of LFs with and without ADSC treatment in two IPF patients’ LFs; genes that were upregulated more than twofold were considered candidate genes for further study. Among the 58201 probes, 317 and 75 genes were upregulated in patients 1 and 2, respectively. The list of the top 30 upregulated genes in both are in Table 1, and an MA plot is shown in Figure 3A. The top 30 upregulated genes were mostly non-coding RNA or cytoskeleton-associated genes. We focused on Protein Tyrosine Phosphatase, Receptor Type R (*PTPRR*) here because it is well known to induce the dephosphorylation of ERK1/2, which plays an important role in the process of fibrotic response; it is also well known that the only drugs used in IPF (pirfenidone and nintedanib) exhibit anti-fibrotic effects by inhibiting tyrosine kinases [18]. We confirmed that the gene expression of *PTPRR* was increased in LFs with ADSC treatment using quantitative PCR (Figure 3B). Moreover, the Western blot analysis demonstrated that the phosphorylation of ERK1/2 was critically inhibited in LFs with ADSC treatment (Figure 3C). ADSCs produce many kinds of cytokines. Hepatocyte growth factor (HGF) is one of the most-produced cytokines from ADSCs, which acts as an anti-fibrotic and anti-inflammatory factor [19,20,21]. Therefore, we investigated whether HGF treatment directly induced the *PTPRR* expression and dephosphorylation of ERK. However, HGF treatment neither increased *PTPRR* expression nor inhibited ERK phosphorylation. These data show that the anti-fibrotic effect of ADSC treatment uses a different system from that of HGF treatment to increase *PTPRR* expression and inhibit phosphorylated ERK (p-ERK).

### 3.3. PTPRR Is One of the Key Regulators of Lung Fibrosis

In order to validate the role of PTPRR, F-LFs transfected with *PTPRR* were analyzed to examine whether *PTPRR* upregulation could suppress fibrotic factors. *PTPRR* overexpression was confirmed through both qPCR and Western blot analysis (Figure 4A). PTPRR overexpression significantly suppressed the expression of fibrosis-related genes *ACTA2*, *COL1A2*, and *COL3A1* (Figure 4B). Furthermore, *PTPRR* overexpression dephosphorylated ERK1/2 (Figure 4C), and the ratios of both p-ERK/β-actin and p-ERK/t-ERK were significantly reduced. These data indicate that *PTPRR* overexpression could suppress fibrotic factors.

### 3.4. ADSCs Improved BLM-Induced Pulmonary Fibrosis (PF) in a Mouse Model

To investigate the anti-fibrotic effect of ADSCs in vivo, we used a well-established PF mouse model affected by BLM. BLM was administrated to mice at 1.5 mg/kg by intratracheal (i.t.) injection; then, 1.5 million ADSCs were transferred intravenously 3 days after BLM administration. The analysis was performed 14 days after BLM administration (Figure 5A). Pathological analysis through HE staining and Masson’s trichrome staining showed the characteristic features of PF, the infiltration of inflammatory cells, the disruption of the alveolar structure, and the deposition of collagen fibers in the control group with the BLM model (BLM i.t. with PBS i.v.), while the ADSC-treated group with the BLM model (BLM i.t. with ADSCs i.v.) improved the pathological features of PF (Figure 5B). The Ashcroft score (Figure 5C) and collagen content in the lung lobe (Figure 5D) were also significantly decreased in the ADSC-treated group. Additionally, the mRNA of fibrotic factors *Col1A2*, *Col3A1*, and *Acta2* was significantly downregulated in the ADSC-treated group (Figure 5E). These data show that ADSC treatment improves the BLM-induced PF model.

### 3.5. ADSCs Improved PF in IPF-Humanized Mice

It has been reported that BLM-induced PF is reversible, and cannot completely mimic the irreversible pathology of IPF [13,22]. To investigate the anti-fibrotic effect of ADSCs on the pathology of IPF, we tested the anti-fibrotic effect of ADSCs on IPF-humanized mice [14]. The pathology of IPF was induced by the intravenous administration of F-LF to SCID-Beige mice. At 35 days after F-LF administration, ADSCs were intravenously transferred to examine the anti-fibrotic effect of ADSCs on the pathology of IPF (Figure 6A). A pathological analysis using Masson’s trichrome staining revealed that collagen deposition was detected in the control group in the humanized IPF model (F-LF i.v. with PBS i.v.), while it was decreased in the ADSC-treated group (F-LF i.v. with ADSCs i.v.) (Figure 6B). The Ashcroft score (Figure 6C) and collagen content in the lung lobe (Figure 6D) were also significantly decreased in the ADSC-treated group compared with the control group. In addition, the mRNA of fibrotic factors *Col1A2* and *Col3A1* was significantly decreased, while *Acta2* tended to decrease without a significant difference. Interestingly, *Ptprr* was upregulated in the ADSC-treated group in the humanized IPF model (Figure 6E). These data indicate that ADSC treatment also improves the pathology of PF in the humanized IPF model.

## 4. Discussion

IPF is characterized by some degree of lung inflammation and abnormal tissue repair. These events consequentially increase collagen gene expression, including collagen type I and α-SMA [23,24]. Currently, only pirfenidone and nintedanib are used to delay lung function decline and cannot extend the lifespan of IPF patients [18]. In fact, there is no definitive treatment for IPF, and an alternative therapy is needed. In this study, we demonstrated the anti-fibrotic effect of ADSCs on LFs from IPF patients through *PTPRR* upregulation, which inhibits ERK1/2 phosphorylation. In addition, we indicated the therapeutic effect of intravenous ADSC administration not only on a BLM-induced pulmonary fibrosis model, but also on a humanized IPF mouse model.

Recent studies have demonstrated that MSCs are important for the cell-based therapy of lung diseases, including IPF, due to their immunomodulatory and regenerative properties, as well as having limited side effects in various experimental animal models. Preclinical studies for IPF have also shown the use of MSCs from BM [25]. Among the MSCs derived from different origins**, however,** ADSCs are favorable because they can be easily isolated from fat tissue collected through liposuction, and the number of collectible cells per volume is higher than that for BM [26,27]. Some studies have shown that ADSCs have anti-inflammatory and anti-fibrotic effects in animal models of BLM-induced lung fibrosis [28,29,30]. We first demonstrated that IPF patient-derived LFs cocultured with ADSCs decreased fibrotic factors, including collagen type I, type III, and α-SMA, and induced *PTPRR* expression, which led to the dephosphorylation of ERK1/2, one of the most important signaling cascades among the MAPK signal transduction pathways [31]. These data are consistent with the previous finding that an ADSC-conditioned medium reduces gene expressions of collagen types I and III as well as α-SMA in fibroblasts via the MAPK signaling pathway [32]. The dephosphorylation of ERK1/2 by PTPRR can regulate downstream MAPK steering cell proliferation, differentiation, and other functions [33]. In particular, LFs from the F area in IPF patients showed a higher induction of *PTPRR* with decreased levels of p-ERK through a coculture with ADSCs. Moreover, the overexpression of *PTPRR* in LFs is sufficient to decrease the phosphorylation of ERK1/2 and reduce fibrotic markers, such as collagen types I and III and α-SMA. These data suggest that PTPRR induction on LFs through coculture with ADSCs leads to decreased fibrotic markers via the dephosphorylation of ERK1/2, which might result in the inhibition of the proliferation, differentiation, and transformation of LFs. ADSCs produce a large number of growth factors, including HGF and a member of the epidermal growth factors. HGF mediated by MSCs had protective effects on an in vivo pulmonary fibrosis model. ADSC administration was also found to recover the injured lung tissue by enhancing the secretion of HGF, and to exert a protective effect by ameliorating inflammation and reducing the degree of lung injury and fibrosis [31]. However, HGF was not enough to induce *PTPRR* and the dephosphorylation of ERK in our study, suggesting that other secretory factors from ADSCs, including other cytokines and extracellular vesicles, might contribute to the induction of *PTPRR* [34]. However, the factor that enhances the PTPRR level is still unknown; thus, further studies are necessary to define the molecular mechanism involved in the promotion of PTPRR in LFs.

Among the currently applied models of experimentally induced fibrosis, the administration of BLM is used most frequently [35]. However, none of the animal models of experimental fibrosis fully recapitulate IPF pathology; thus, recent attempts have been made to “humanize” the models. The best-characterized humanized model involves the intravenous administration of human IPF fibroblasts into immunodeficient SCID-Beige mice [13]. These mice lack many features of innate and acquired immune responses, allowing human cells to be grown in the lung. This model has enabled important observations to be made regarding the pathogenic potential of IPF fibroblasts. Interestingly, although human fibroblasts directly contribute to the pathologic remodeling in mouse lungs, these cells also activate murine epithelial cells and fibroblasts to enable pathologic remodeling and proliferation. In one study, mice exhibited significant PF histopathology, which was evident at 35 days after the i.v. injection of IPF fibroblasts [36]. In this study, ADSCs were intravenously administered at day 35 after the i.v. injection of IPF fibroblasts to investigate the therapeutic effect of ADSCs against lung fibrosis in vivo, and the findings showed that the intravenous administration of ADSCs improved the pathogenesis of lung fibrosis, indicating that ADSCs could indirectly regulate the remodeling and proliferation of IPF fibroblasts in vivo through secretory factors. However, this model has the disadvantage of modeling fibrosis development in the absence of immune cells, which does not occur in humans.

We failed to determine whether ADSCs were delivered to the lung fibroblasts in this study. However, we found that i.v. injection can deliver not only ADSCs, but also secretary molecules, including cytokines and exosomes, contributing to the anti-fibrotic effect on lung fibroblasts. In addition, we previously reported that ADSCs ameliorate elastase-induced emphysema in mice [37]. In the study, we used PKH-labeled ADSCs to trace the cells and determine whether intravenously injected ADSCs can migrate into lung tissue. We detected PKH-labeled ADSCs in the lung at 25 days after i.v. injection. In this study, we focused on the secretory factors from ADSCs resulting from coculturing with ADSCs and LFs. However, further studies are necessary to measure the effects of i.v.-delivered ADSCs on lung fibroblasts more accurately.

## 5. Conclusions

This is the first demonstration of the anti-fibrotic effect of ADSCs on LFs from IPF patients both in vitro and in vivo. These results suggest that ADSC therapy might be a potential therapeutic approach for IPF patients.

## Figures and Tables

**Figure 1 cells-13-02050-f001:**
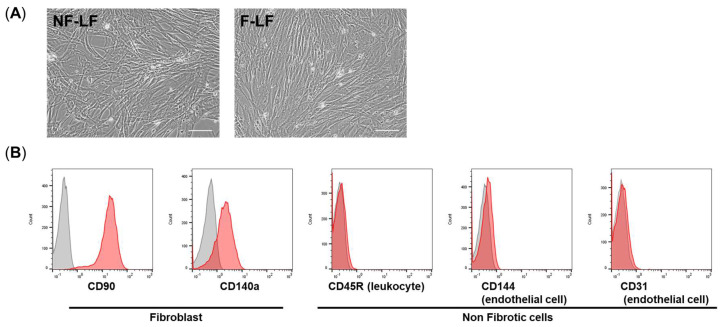
Characterization of LFs isolated from IPF patients. LFs were isolated from both areas of human lung samples in accordance with a previously described procedure [16]. The characterization of our LFs is shown in (**A**), a micrograph of NF-LF and F-LF, and in (**B**), an analysis of the expression of markers of fibrotic cells (CD140a and CD90), leukocytes (CD45), and endothelial cells (CD31 and CD144) using flow cytometry. The gray area is the result of LFs without Ab staining, and the red area is the result of LFs with Ab staining. Scale bars: 100 µm.

**Figure 2 cells-13-02050-f002:**
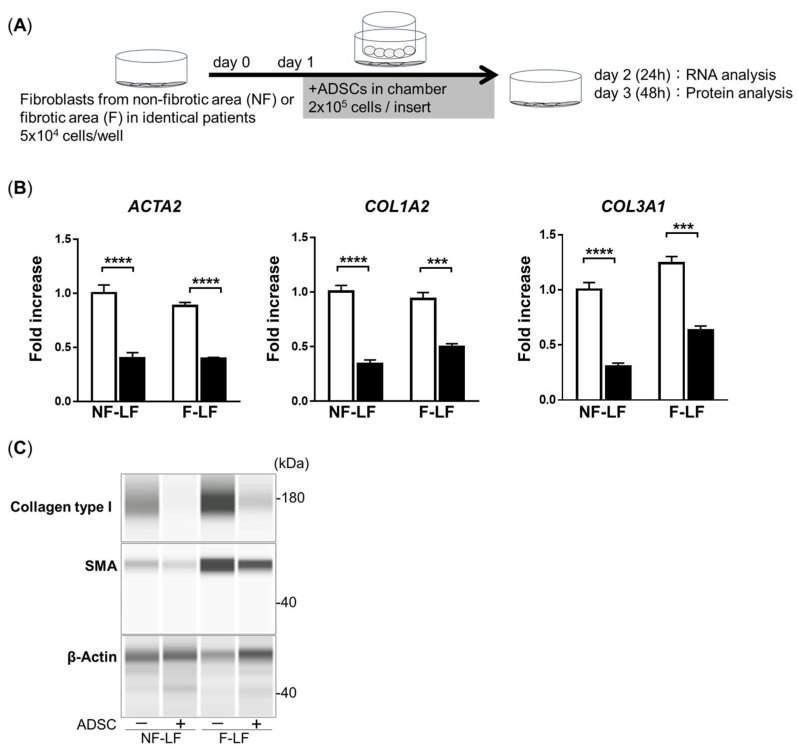
ADSC treatment inhibits the expression of fibrotic factors in fibroblasts derived from IPF patients. (**A**) The methodology of coculture with ADSCs and LFs. (**B**) Expressions of *ACTA2*, *COL1A2*, and *COL3A1* in LFs with (black bars) or without (white bars) ADSC treatment at 24 h. *GAPDH* was used as an internal control. Data are represented by the mean ± SEM from three independent experiments. *p*-value; **** *p* < 0.0001, *** *p* < 0.001. (**C**) The protein expression of α-SMA and collagen type I standardized using β-actin at 48 h.

**Figure 3 cells-13-02050-f003:**
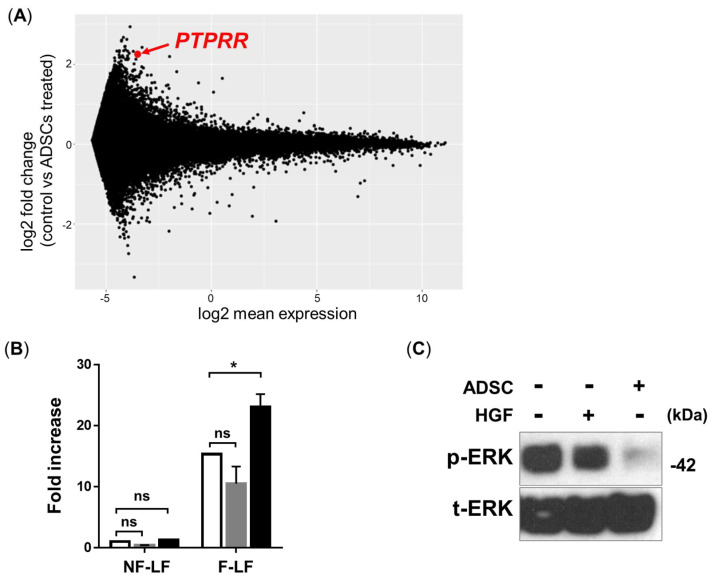
ADSC treatment upregulates the expression of *PTPRR* in LFs. (**A**) The results of the DNA microarray analysis of LFs with ADSC treatment are shown as an MA plot. The red dot indicates *PTPRR* expression. The data include the gene expression of F-LFs derived from two IPF patients. (**B**) qPCR analysis of *PTPRR* gene expression of LFs stimulated with ADSCs (black bars) or HGF (gray bars). The white bars show the data of non-treated NF-LFs or F-LFs. The fold increase in *PTPRR* expression was calculated using the *PTPRR* expression of non-treated NF-LFs. Data are represented as the mean ± SEM from three independent experiments. *p*-value; * *p* < 0.05., ns; not significant. (**C**) Western blotting analysis of p-ERK/t-ERK expression in LFs stimulated with ADSCs or HGF.

**Figure 4 cells-13-02050-f004:**
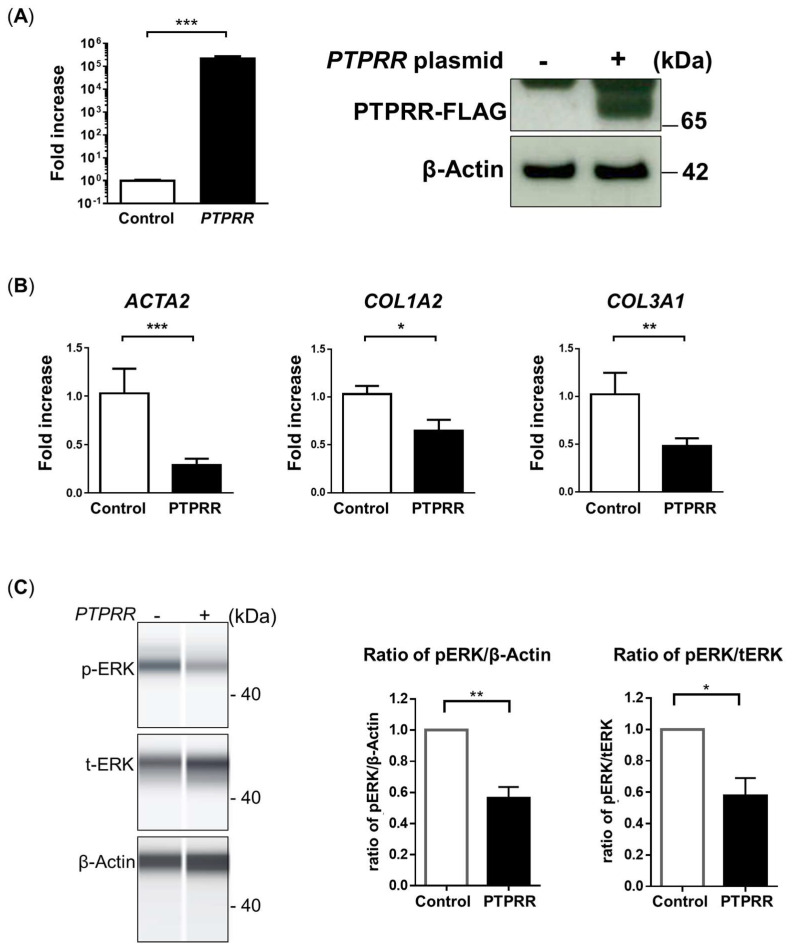
Overexpression of *PTPRR* in LFs suppressed fibrotic factors. (**A**) qPCR analysis of *PTPRR* gene expression of LFs transfected by control or *PTPRR* plasmid (left panel). FLAG-tagged PTPRR protein expression in *PTPRR*-transfected LFs was detected using anti-FLAG monoclonal antibody (right panel). (**B**) qPCR analysis of fibrotic factors *ACTA2*, *COL1A2*, and *COL3A1* tested in *PTPRR*-transfected LFs. (**C**) Protein analysis of p-ERK/t-ERK in *PTPRR*-transfected LFs via the WES system. The expression levels of each protein were normalized on GAPDH or t-ERK protein levels, respectively, and ratios for both p-ERK/β-actin and p-ERK/t-ERK were calculated. Data are represented by the mean ± SEM from three independent experiments. *p*-value; *** *p* < 0.001, ** *p* < 0.01, * *p* < 0.05.

**Figure 5 cells-13-02050-f005:**
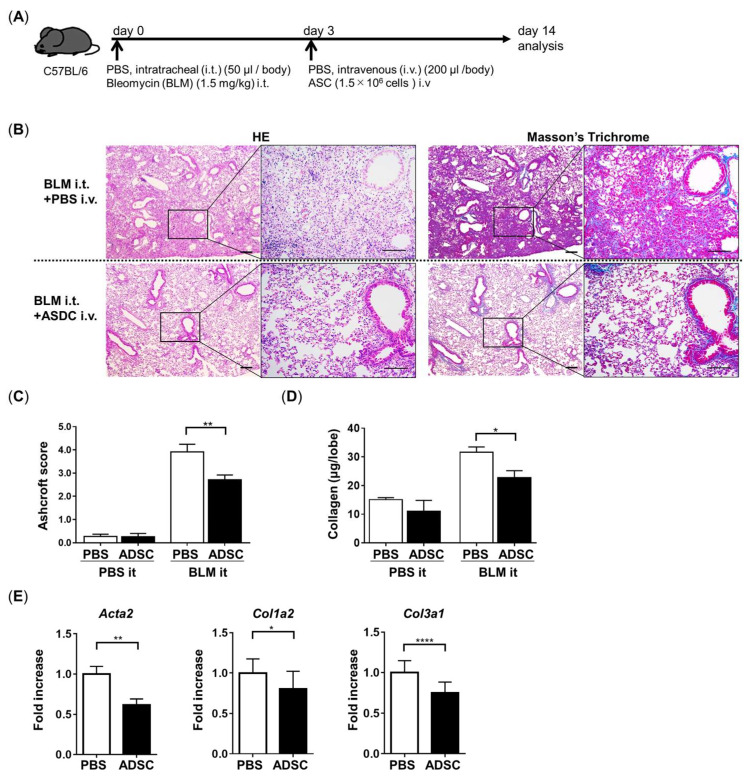
Treatment of ADSCs improved BLM-induced PF. (**A**) Time course of this experiment: C57BL/6 mice were treated with BLM intratracheally and then administered with ADSCs intravenously at day 3. The mice were euthanized on day 14 for pathological analysis (HE staining and Masson’s trichrome staining) (**B**), Ashcroft scores indicating the level of fibrosis (**C**) and collagen content (**D**). (**E**) qPCR of fibrotic factors *Acta2, Col1a2*, and *Col3a1* was performed in a lobe of lung tissue. Scale bars: 100 µm. Data are represented by the mean ± SEM from three independent experiments. *p*-value; **** *p* < 0.0001, ** *p* < 0.01, * *p* < 0.05.

**Figure 6 cells-13-02050-f006:**
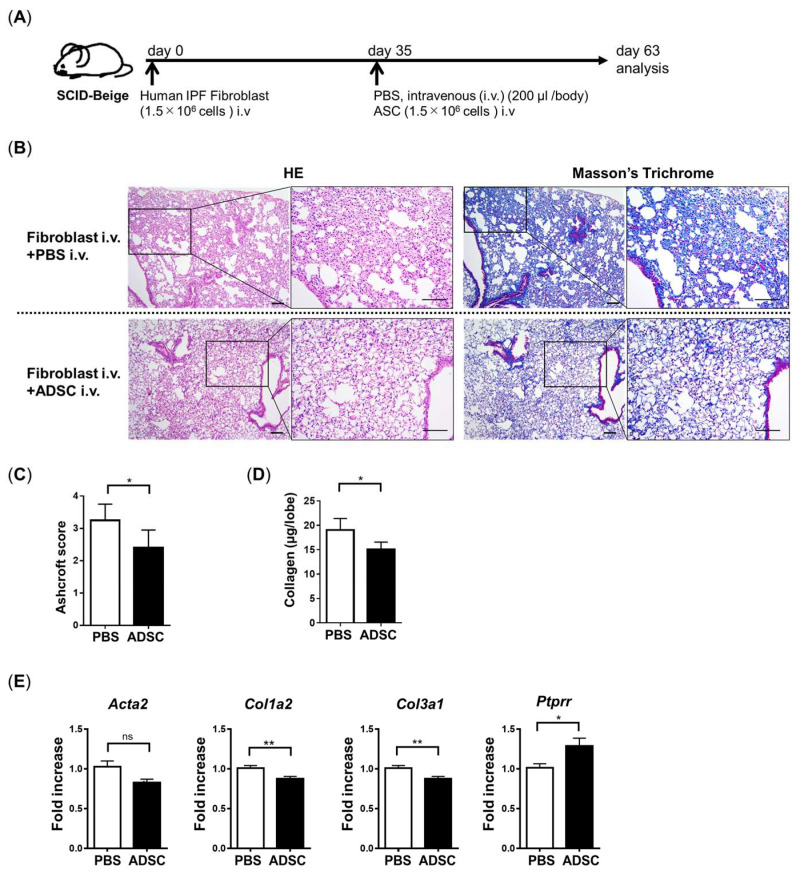
Treatment with ADSCs improved PF in the IPF-humanized mouse model. (**A**) Time course of this experiment: SCID-Beige mice were treated with F-LFs intratracheally and then administrated with ADSCs intravenously at day 35. The mice were euthanized on day 63 for pathological analysis (HE staining and Masson’s trichrome staining) (**B**), Ashcroft scores indicating the level of fibrosis (**C**) and collagen content (**D**). (**E**) Quantitative PCR of fibrotic factors *Acta2, Col1a2*, *Col3a1*, and *Ptprr* performed in a sample of lobe lung tissue. Scale bars: 100 µm. Data are represented by the mean ± SEM from three independent experiments. ** *p* < 0.01, * *p* < 0.05. ns; not significant.

**Table 1 cells-13-02050-t001:** Top 30 genes upregulated in LFs treated with ADSCs.

	GeneSymbol	log2 Fold Change (Control vs. ADSCs Treated)	log2 Mean Expression
1	*ZNF91*	2.9433	−3.8617
2	*C10orf99*	2.6767	−4.1783
3	*LINC01115*	2.6267	−4.0167
4	*NA*	2.6100	−4.2617
5	*lnc-NPHS2-1*	2.5833	−4.2717
6	*NA*	2.4533	−4.1000
7	*DKFZP547J0410*	2.4467	−4.1033
8	*ABRA*	2.4467	−4.1133
9	*TMEM71*	2.4267	−3.2933
10	*NA*	2.3667	−4.0800
11	*LOC283140*	2.3433	−4.1650
12	*PRAC1*	2.3433	−4.3683
13	*MYH8*	2.3333	−4.4900
14	*LINC01057*	2.3333	−3.9800
15	*DNAH17*	2.3267	−3.7100
16	*lnc-CHAC1-3*	2.3167	−4.3750
17	*lnc-SLC25A26-1*	2.2800	−3.9033
18	*PTPRR*	2.2667	−3.5133
19	*lnc-TMEM64-3*	2.2400	−4.3833
20	*CCNI2*	2.2167	−4.2217
21	*SLC7A11-AS1*	2.2133	−3.9800
22	*lnc-ANTXRL-2*	2.1967	−1.9883
23	*lnc-AC079341.1-2*	2.1933	−4.2367
24	*ZNF252P-AS1*	2.1800	−4.3233
25	*XLOC_l2_009332*	2.1733	−4.3700
26	*DLG2*	2.1733	−4.1867
27	*lnc-PHF3-2*	2.1633	−4.5983
28	*MPL*	2.1500	−4.0050
29	*AMER1*	2.1433	−3.9417
30	*SNORD19*	2.1300	−3.5850

## Data Availability

The datasets used and/or analyzed during this study are available upon reasonable request from the corresponding author.

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
