# Peer review of "Adipose-Derived Mesenchymal Stem Cells (ADSCs) Have Anti-Fibrotic Effects on Lung Fibroblasts from Idiopathic Pulmonary Fibrosis (IPF) Patients"

_cells, 2024, doi:10.3390/cells13242050_

Round 1
Reviewer 1 Report
Comments and Suggestions for Authors
Adipose tissue-derived MSCs (ASCs) are regarded as a promising resource for cell-based therapies. The present manuscript illustrates the in vitro efficacy of these cells in atenuating the signs of fibrosis in lung fibroblasts derived from patients with IPF and in vivo in lung fibrosis severity following ASCs administration in two mouse models.
Specific comments
A brief description of the isolation of LFs from lung tissue and the characterization of cultured LFs must be included. It is unclear how the authors have distinguished LFs from other stromal cells present in lung tissue, such as MSCs, pericytes, and SMCs.
The primary or passaged LFs were used?
The rationale behind the relatively brief LF cultivation and LF-ASC co-cultivation periods is unclear. It appears that a period of 24 hours is insufficient for the cells to adapt to the culture conditions subsequent to plating.
The authors have employed PCR exclusively for the purpose of characterizing the effects of ASCs. The transcriptional alterations are not sufficient for evaluating the anti-fibrotic processes.
In applying PCR analysis to lung tissue, it is unclear how the authors could have been aware that they were measuring the effects of i.v. delivered ASCs on lung fibroblasts exactly?
Author Response
Adipose tissue-derived MSCs (ASCs) are regarded as a promising resource for cell-based therapies. The present manuscript illustrates the in vitro efficacy of these cells in attenuating the signs of fibrosis in lung fibroblasts derived from patients with IPF and in vivo in lung fibrosis severity following ASCs administration in two mouse models.
Specific comments
A brief description of the isolation of LFs from lung tissue and the characterization of cultured LFs must be included. It is unclear how the authors have distinguished LFs from other stromal cells present in lung tissue, such as MSCs, pericytes, and SMCs.
→Thank you for pointing it out. We added a brief description of the isolation of LFs from lung tissue in Materials and Methods (Line 80), and added data about the characterization of cultured LFs in Materials and Methods (Line 87, Figure 1).
The primary or passaged LFs were used?
→ We apologize for confusing. We used passaged LFs in this experiment.
We added the explanation about LFs we used in Materials and Methods (Line 90).
The rationale behind the relatively brief LF cultivation and LF-ASC co-cultivation periods is unclear. It appears that a period of 24 hours is insufficient for the cells to adapt to the culture conditions subsequent to plating.
→ We performed the experiment of the co-culture system at 24 and 48 h, and got results with the same tendency on the co-culture system both at 24 h and 48 h.
We added the data about the protein analysis at 48h and used the data about RNA analysis at 24 hours (Figure 2), because the periods could show our result more clearly.
The authors have employed PCR exclusively for the purpose of characterizing the effects of ASCs. The transcriptional alterations are not sufficient for evaluating the anti-fibrotic processes.
In applying PCR analysis to lung tissue, it is unclear how the authors could have been aware that they were measuring the effects of i.v. delivered ASCs on lung fibroblasts exactly?
→ We added the data about the protein analysis at 48h on co-culture system to show the anti-fibrotic effect of ADSCs by indirect association with fibroblast more clearly.
We do not see whether ADSCs was delivered to lung fibrosis in this study, and we consider that not only i.v. delivered ADSCs but also secretary molecules, including cytokines and exosomes, from ADSCs contribute to anti-fibrotic effect on lung fibroblast.
Also, we have previously reported that ADSCs have ameliorated elastase-induced emphysema in Mice (Int J Chron Obstruct Pulmon Dis. 2021 Oct 8;16:2783–2793. doi: 10.2147/COPD.S324952, reference 16). In the study, we used PKH-labeled ADSCs to trace ADSCs whether intravenously injected ADSCs could migrate into lung tissue. We have detected PKH-labeled ADSCs in lung at 25 days after i.v. injection of ADSCs. However, further studies are necessary to measure the effects of i.v. delivered ADSCs on lung fibroblasts exactly. We added it to the Discussion (Line 404).
Reviewer 2 Report
Comments and Suggestions for Authors
Good manuscript with only a few issues.
The data for RNAseq should not be in supplemental but should be in the manuscript.
Were all 30 genes that had more than 2-fold increase considered as candidates for further study? A better description of why PTPRR was selected for further study out of the 30 genes listed needs to be provided. There are several other good candidates for consideration.
The histology data in Figures 4 and 5 were compelling, but I really wanted to see a higher power view. Please provide higher power image so that cell structure and specific areas that were stained can be seen/examined.
It is not just seeing an increase in stain color in an image but seeing that there is increased collagen or decreased collagen in a higher power image of the cells and collagen that is important.
Comments on the Quality of English LanguageLine 195 and 196 states "then, we picked up the gene that was up-regulated by more than 2-fold should be rewritten. This should state that genes that were up-regulated more than 2-fold were considered as candidate genes for further study.
There were a few areas like this that need to be rewritten.
Author Response
Good manuscript with only a few issues.
The data for RNAseq should not be in supplemental but should be in the manuscript.
→ We transfer the supplemental data to in the manuscript (Table 1).
Were all 30 genes that had more than 2-fold increase considered as candidates for further study? A better description of why PTPRR was selected for further study out of the 30 genes listed needs to be provided. There are several other good candidates for consideration.
→ Thank you for pointing it out. It could be considered that all 30 genes that had more than 2-fold increase have considered as candidates for cell-therapy of IPFs. We added the description of why PTPRR was selected and transferred the supplemental data to in the manuscript (Line 235).
The histology data in Figures 4 and 5 were compelling, but I really wanted to see a higher power view. Please provide higher power image so that cell structure and specific areas that were stained can be seen/examined.
It is not just seeing an increase in stain color in an image but seeing that there is increased collagen or decreased collagen in a higher power image of the cells and collagen that is important.
→ Thank you for pointing it out. We added the higher power image in Figure 5 and 6.
Line 195 and 196 states "then, we picked up the gene that was up-regulated by more than 2-fold should be rewritten. This should state that genes that were up-regulated more than 2-fold were considered as candidate genes for further study.
There were a few areas like this that need to be rewritten.
→ Thank you for pointing it out. We have corrected it as you pointed out (Line 228).
Round 2
Reviewer 1 Report
Comments and Suggestions for Authors
The authors have thoughtfully addressed the issues raised by the reviewer. Responses and changes in the text are accepted by the reviewer as satisfactory